# GENPLAN

## ABSTRACT

We present GenPlan, a novel deep learning architecture for generating architectural floor plans. GenPlan provides flexibility and precision in room placement, offering architects and developers new avenues for creative exploration. We adapted an autoencoder-like structure comprising of two encoders and four specialized decoders that predict the centers of different rooms. These predictions are converted into graph along with the other constraints and used as inputs for a Transformer-based graph neural network (GNN), which is responsible for delineating room boundaries and refining the predicted room centers. The Graph Transformer Network ensures that the generated floor plans are realistic and executable in real-life. GenPlan's methodological innovation provides heightened control during the design phase, serving as a valuable tool for automating and refining the architectural design process.

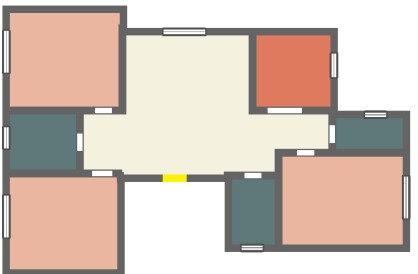 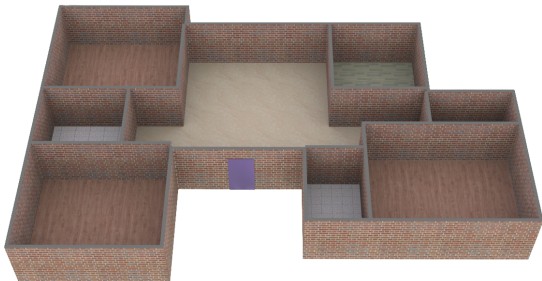

(a) Detailed 2D Architectural Floor Plan given only the shape on the land and the front door position as inputs

(b) Automated Textured 3D Model of the Generated Floor Plan using GenPlan

Figure 1: GenPlan generates floor plans as those shown in (a) which can then be rendered to a fully automated 3D model as shown in (b).

## 1 INTRODUCTION

The generation of architectural floor plans is a foundational step in building design, directly impacting not only aesthetics but also the functionality and sustainability of living spaces. Traditionally, this process has been time-consuming and prone to human error, particularly as design complexity increases. However, recent advancements in computational design, especially through deep learning, have begun to transform this landscape. Research by Chaillou (2020), Hu et al. (2020), Sun et al. (2022), Wang et al. (2023), Upadhyay et al. (2023), demonstrates the effectiveness of generative adversarial networks (GANs) and deep learning models in automating and optimizing layout generation. These advanced methods eliminate the need for brute-force strategies and traditional optimization algorithms, which often struggle to find solutions in a reasonable timeframe. Moreover, traditional approaches face challenges in achieving optimal metrics, such as golden ratios in spatial design. This limitation further highlights the superiority of deep learning models in efficiently generating optimized designs.

Moreover, most prior work has incorrectly classified balconies as part of the interior area. In fact, balconies are exterior spaces and should not be included in the carpet area or floor area Chow (2002).

Considering the balcony as part of the floor area not only misrepresents the true use of space but also limits the design process, as architects typically do not include balconies in the carpet area when designing a floor plan.

Unlike previous methods that primarily focus on generating basic layouts, GenPlan integrates state-of-the-art techniques from both convolutional neural networks (CNNs) and graph neural networks (GNNs) in a sequential manner to ensure conflict-free designs.

GenPlan consists of two sequential modules: GenCenter and GenLayout. GenCenter utilizes shared ResNet101 encoders and specialized decoders to predict the locations of various room types, including bedrooms, restrooms, and kitchens, based on the given floor plan boundary. These predictions are then converted into a graph structure that incorporates additional constraints, which serves as input for GenLayout, our Transformer-based Graph Neural Network (GNN). This module is responsible for delineating room boundaries and refining the predicted room centers.

The GenLayout module ensures that the generated floor plans are both realistic and executable in real-world scenarios. By innovating this methodological approach, GenPlan enhances control during the design phase, positioning itself as a valuable tool for automating and refining the architectural design process.

To illustrate GenPlan's functionality, consider an architect exploring specific design ideas. They input various constraints, and GenPlan generates multiple layouts with different room counts and configurations. Additionally, the architect can interact with the system during the process, modifying aspects like room location or removing rooms entirely, offering a more flexible and adaptable interactive design experience.

## 2 RELATED WORK

Researchers have extensively applied deep learning models, such as generative adversarial networks (GANs) and variational autoencoders (VAEs) to generate floor plans based on the shape of the floor and the position of the front door, as demonstrated by Sun et al. (2022), Chaillou (2020), Hu et al. (2020), Nauata et al. (2021), Wang et al. (2023), Upadhyay et al. (2023), Chen (2022), Ślusarczyk et al. (2023), Huang & Zheng (2018). However, these models often face challenges in adhering to specific architectural standards or functional requirements. For instance, many previous works incorrectly include balconies as part of the interior area, misclassifying them as floor or carpet areas. This approach limits the design process, as architects typically do not consider balconies in the carpet area when designing a floor plan.

This area of research faces several challenges, such as the need for extensive and diverse datasets to train more robust models and the integration of user-specific preferences and constraints. Additionally, the development of comprehensive evaluation metrics that effectively assess the usability and quality of generated designs remains crucial.

## 3 METHODOLOGY

Once a room center is predicted, a Transformer-Convolutional GNN is employed to predict the rectangular dimensions of the room. This is done using the boundary polygon, front door position, and the predicted room centers from the first phase; the GNN processes this information and outputs the appropriate dimensions for each room, ensuring they fit within the overall floor plan layout and respect spatial relationships between adjacent rooms and boundaries.

After the room layout is generated, doors and windows are placed using a similar architecture to the one used for generating room centers, but this time they are trained specifically to generate the positions of doors and windows. Then, all elements—walls, rooms, doors, and windows—are overlaid on top of each other. We apply geometric operations to adjust the layout, including cutting out parts of the room rectangles that extend beyond the floor boundary. This process can result in non-rectangular room shapes similar to those commonly seen in real-world floor plans.

### 3.1 DATA PREPARATION

In the development of GenPlan, we created the ResPlan dataset, which comprises 17,000 diverse real-world floor plans sourced from various real estate websites. These floor plans are represented as geometric shapes, which not only significantly reduce data size but also facilitate the seamless conversion of floor plans into various formats, such as graphs and images. By addressing common dataset challenges, such as overly simplistic floor plans and a lack of diversity—where most designs in the RPlan dataset features only a single restroom, and it is rare to encounter a floor plan with three restrooms or more than three bedrooms, as well as issues with incorrect room connections—the ResPlan dataset not only enhances the training process for GenPlan but also establishes a standard for comparison in this field.

For the convolutional neural network (CNN), the floor plan boundaries and front door positions are converted into binary masks (see Figure 3). These inputs are critical as they define the minimum constraints required to generate a floor plan. Other inputs, such as room counts, are encoded as a one-hot encoded 8-channel image representing up to four bedrooms and four bathrooms, which are the limits of our dataset and align with most residential floor plan requirements. The area is encoded in a single channel as a solid binary square, where its length is linearly scaled from 0 to 255, corresponding to a range of 20 to 400 square meters. These inputs are concatenated into an 11-channel input for one of the two encoders used in the room center generation component.

For the graph input version, all elements are converted into a graph structure. The boundary is represented using nodes for corners and edges for walls. The front door is added as a node that intersects one of the wall edges, with each node encoded using one-hot encoding based on its type. After generating the room centers, they are also included in the graph as nodes, with distinct one-hot encodings corresponding to room types. Each room node is then connected to the five closest boundary corner nodes with edges. Additionally, all room center nodes are interconnected to facilitate learning about each other's positions and to avoid conflicts in acquiring areas.

### 3.2 INITIAL ROOM COUNT PREDICTION

A preliminary convolutional neural network (CNN), specifically a pre-trained ResNet18, is utilized to analyze the boundary, front door position, and the total area of the plot to estimate the required number of bedrooms and restrooms. This model leverages historical data and trained parameters to predict an optimal distribution of rooms based on spatial dimensions and entry points. The output guides the spatial distribution within the floor plan:

$$N_{counts} = \text{RoomCounter}(\text{Boundary}, \text{Front Door Position}, \text{Area}; \theta_{counts}) \qquad (1)$$

where $N_{rooms}$ is the suggested number of bedrooms and restrooms that could fit in this floor area, Boundary is the the floor plan boundary mask, Front Door Position is the designated entry point, and Area is the encoded area of the floor plan boundary.

Moreover, while the model provides an automated estimate, users can override these suggestions to exert greater control over the design. This feature is crucial for accommodating specific client requests or adapting to unique architectural challenges, allowing designers to adjust the number of Bedrooms and Restrooms, and hence providing greater diversity in design outcomes.

### 3.3 ROOM CENTER GENERATION (GENCENTER)

After the initial room count prediction (which can be skipped by inputting the desigred bedrooms and restrooms number), the GenPlan architecture employs a shared ResNet101 encoder to process the floor plan boundary front door and roos numbers encoded as one hot and extract essential features for predicting the center coordinates of various room types. This shared encoder architecture significantly reduces computational redundancy and enhances consistency across the output from the specialized decoders. The use of a shared encoder ensures that the feature extraction process is uniform and only needs to be executed once, thus speeding up the prediction process:

$$F_{shared} = \text{LayoutEncoder}(\text{Boundary}, \text{Front Door}, \text{Room Count}, \text{Area}, F_{recurrent}; \theta_{shared}) \qquad (2)$$

where $F_{\text{recurrent}}$ represents compressed features from previous predictions, encapsulating essential spatial and structural information, and $\theta_{\text{shared}}$ are the parameters of the ResNet101 model.

## 3.4 TWO ENCODER FOR CONTEXTUAL FEATURE COMPRESSION

Following the decoding process, a recurrent encoder compresses the images depicting the detected room centers into a compact feature representation. These images, illustrating centers as circles with a 5-pixel radius, are transformed into feature maps of dimensions $8 \times 8 \times 512$. This compression is crucial for preparing the features for iterative decoding, allowing each subsequent decoder to be informed by previously predicted centers:

$$F_{\text{recurrent}} = \text{RecurrentEncoder}(\text{Centers}, \text{Boundary}, \text{Front Door}, \text{Area}; \theta_{\text{recurrent}}) \quad (3)$$

where $F_{\text{recurrent}}$ represents the compressed feature vector derived from the detected room centers, Centers represent 4 channel binary image containing the locations of the predicted room centers so far, and $\theta_{\text{recurrent}}$ are the parameters of the recurrent encoder, tasked with reducing the dimensionality of the input while preserving essential spatial characteristics for further decoding processes. These features are concatenated with the bottleneck features from the shared encoder ($F_{\text{shared}}$) and used as inputs for the specialized decoders for subsequent room type predictions.

To generate the room centers, we employ four specialized decoders; each specialized decoder is responsible for predicting the center coordinates for specific room types—bedrooms, restrooms, kitchens, and balconies. This modular approach allows each decoder to be fine-tuned to recognize distinct features relevant to each room type, such as size and location, increasing the relevancy of the predictions.

$$F_{\text{shared}} = \text{Concat}(F_{\text{layout}}, F_{\text{recurrent}}) \quad (4)$$

$$C_{\text{type}} = \text{Decoder}_{\text{type}}(F_{\text{shared}}; \phi_{\text{type}}) \quad (5)$$

where $C_{\text{type}}$ denotes the predicted single-channel image containing suggestions for the centers of target rooms, an example of which is shown in Figure 2, and $\phi_{\text{type}}$ are the parameters of the decoder specialized for that particular room type, $F_{\text{shared}}$ is the concatenated features from the two encoders.

Initially, the ResNet101-based Layout Encoder was trained with a single general decoder to optimize the extraction of universal features beneficial across all room types. Post this phase, the encoder parameters, alongside those of the recurrent encoder, were frozen to ensure stability and consistency in feature representation. Subsequently, multiple decoders were individually trained using these stable, pre-extracted features. This method not only streamlines the training process but also ensures each decoder is highly specialized for its designated task, although our current focus remains on the essential floor plan elements.

## 3.5 BLOB DETECTION FOR CENTER SEGMENTATION

After obtaining the single-channel image outputs (Figure 2) from the specialized decoders, we apply the Laplacian of Gaussian (LoG) blob detection technique to identify potential room centers. This method effectively distinguishes multiple blobs in the output images, with each blob representing a potential room center. While all detected blobs are valid candidates, we select the one with the highest intensity as the predicted center. This selection is crucial because, although nearly all these blobs could serve as valid room centers, they are not necessarily designed to coexist in the same layout. Therefore, we choose only one blob to ensure clarity in the design.

Once we select a center, we feed it back into the prediction system along with the other generated centers. This iterative process allows the model to determine the most suitable location for the next center based on the context of the existing design. Moreover, by choosing a random center from the produced blobs, we can generate numerous valid designs. Each of these variations remains architecturally sound, as demonstrated in Figure 4.



Figure 2: This is a sample output from GenCenter before and after the blob detection process. The multiple blobs represent potential valid room locations, each with varying intensity levels. Selecting any of these blobs would lead to a different sequence of predictions, and processing each one would yield different room layouts. To optimize the design, we choose the blob with the highest intensity, representing the most likely or optimal room location.

### 3.6 GRAPH-BASED ROOM BOUNDARY DETERMINATION (GENLAYOUT)

This crucial stage in the GenPlan system involves using GenLayout, a graph neural networks (GNN) to precisely delineate and refine room boundaries. The method leverages a composite representation of architectural elements, emphasizing the dynamic integration of spatial data and the adaptability of GNNs to complex layouts. After generating the room centers $(x, y)$ using 3.3, the Graph Transformer Network is used to determine the actual room shape by predicting the diagonal coordinates $(x_1, y_1, x_2, y_2)$, which are then used to form a rectangle, representing the room's shape.

The input graph 3.1 is processed using a GNN that incorporates Transformer Convolution layers. The graph transformer operator used in this work follows the approach from Shi et al. (2020), and implemented by Fey & Lenssen (2019):

$$\mathbf{x}'_i = \mathbf{W}_1 \mathbf{x}_i + \sum_{j \in \mathcal{N}(i)} \alpha_{i,j} \mathbf{W}_2 \mathbf{x}_j, \tag{6}$$

Where the attention coefficients $\alpha_{i,j}$ are computed via multi-head dot product attention:

$$\alpha_{i,j} = \text{softmax} \left( \frac{(\mathbf{W}_3 \mathbf{x}_i)^\top (\mathbf{W}_4 \mathbf{x}_j)}{\sqrt{d}} \right), \tag{7}$$

Where $\mathbf{x}'_i$ is the updated node feature, $\mathbf{x}_i$ and $\mathbf{x}_j$ are the input features of nodes $i$ and $j$, $\mathcal{N}(i)$ denotes the set of neighbors of node $i$, $\mathbf{W}_1$, $\mathbf{W}_2$, $\mathbf{W}_3$, and $\mathbf{W}_4$ are learnable weights, and $d$ is the dimension of the transformed feature space.

Graph Transformer Convolution is chosen for its advanced capabilities in handling spatial data, which is particularly beneficial in the context of floor plan design. The reasons include:

a) Global Receptive Field: Graph Transformer Convolution layers utilize self-attention mechanisms that allow for processing information on a global scale. This capability is crucial for understanding the entire layout of complex floor plans, as it enables the model to consider how different parts of the plan interact with each other.

b) Dynamic Weight Adjustment: These layers can dynamically adjust their weights based on the context provided by the nodes within the graph. This adaptability is essential for accurately modeling the intricate spatial relationships needed for precise delineation of room boundaries, thereby enhancing the effectiveness of the floor plan design.

c) Multi-Head Attention: Although not explicitly shown in the equations, multi-head attention typically involves computing multiple sets of attention coefficients and aggregating the results. This allows the model to capture diverse aspects of the spatial relationships between nodes.

This detailed implementation of Graph Transformer Convolution in GenPlan illustrates its capability to enhance spatial data processing, ensuring that the generated floor plans are optimized for functionality and practicality.

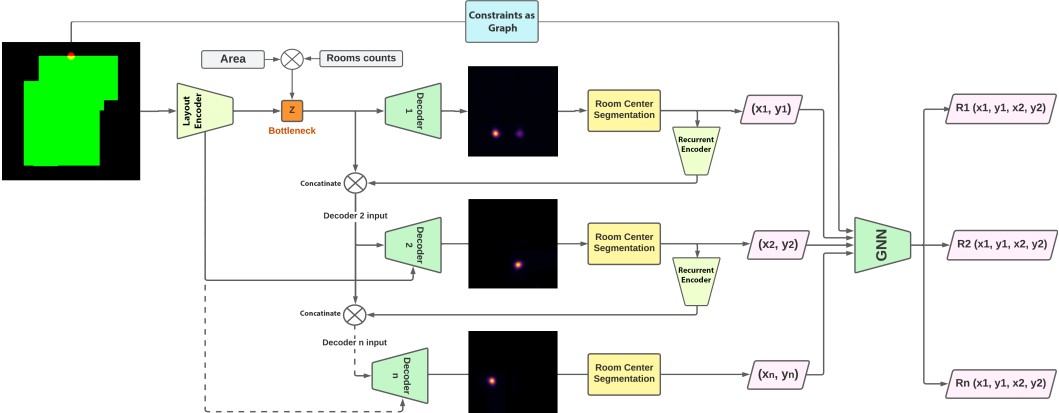

Figure 3: Diagram of the GenPlan architecture, showing the flow of data. Inputs include boundary and front door as binary masks, area, and room counts, which are processed through the CNN and transformer-based GNN parts of GenPlan, generating the final room layout.

## 3.7 TRAINING

GenPlan was trained in two phases, employing Mean Squared Error (MSE) loss, as the task is inherently regression-based. The Adam optimizer was utilized with a batch size of 20 and an initial learning rate of $10^{-2}$, which decayed by a factor of 0.1 every 30 epochs, reaching a minimum learning rate of $10^{-4}$.

In the first phase, we trained the GenCenter to simultaneously generate centers for all room types. Random samples and tasks were input into the model, ensuring that each batch contained examples from all tasks. This phase spanned a duration of 150 epochs.

In the second phase, we replicated the decoder four times, with each instance dedicated to a specific room type: bedrooms, restrooms, kitchens, and balconies. After freezing the parameters of the two encoders, each decoder was trained independently for 50 epochs on its corresponding task.

A similar architecture was employed for the placement of doors and windows, utilizing only two specialized decoders. This distinction was crucial to ensure the precise positioning of doors and windows, as their locations depend on the room boundaries established earlier. The door and window model was trained using the same two-phase strategy as the primary architecture.

We trained GenLayout, a Transformer-based convolutional GNN, on the graph representation of the input constraints, applying various augmentation methods to slightly shift the centers of rooms. This enabled the model to learn to adjust the center of the generated rectangle if it deviates from the optimal position. GenLayout was trained for 300 epochs, using the same learning rate and batch size as GenCenter.

## 3.8 POST-PROCESSING

The final layout is assembled by integrating room boundaries within the plot. This process involves aligning and refining boundaries to create clear partitions. Geometric buffering, a key technique used here, extends or contracts room boundaries by a predefined wall width to model the physical dimensions of walls, ensuring clear delineation and structural integrity.

The algorithm sets a predefined wall width and adjusts the floor plan layout to accommodate wall thickness using geometric buffering. Rooms are sorted by size, and each is buffered to prevent overlaps, maintaining clear boundaries. The main living area is adjusted similarly, optimizing space usage. Leftover spaces are efficiently integrated back into the layout. Finally, room positions are finalized, and visible wall lines are defined based on buffered placements, ensuring a precise and functional architectural layout. This approach effectively combines advanced machine learning with architectural principles, enhancing both the functional aspects of the design.

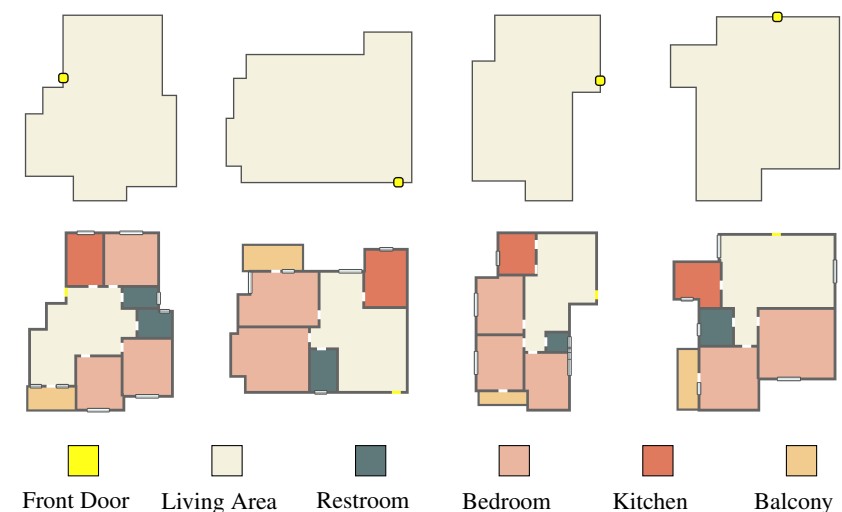

Front Door     Living Area     Restroom     Bedroom     Kitchen     Balcony

Figure 5: Examples of generated floor plans using GenPlan

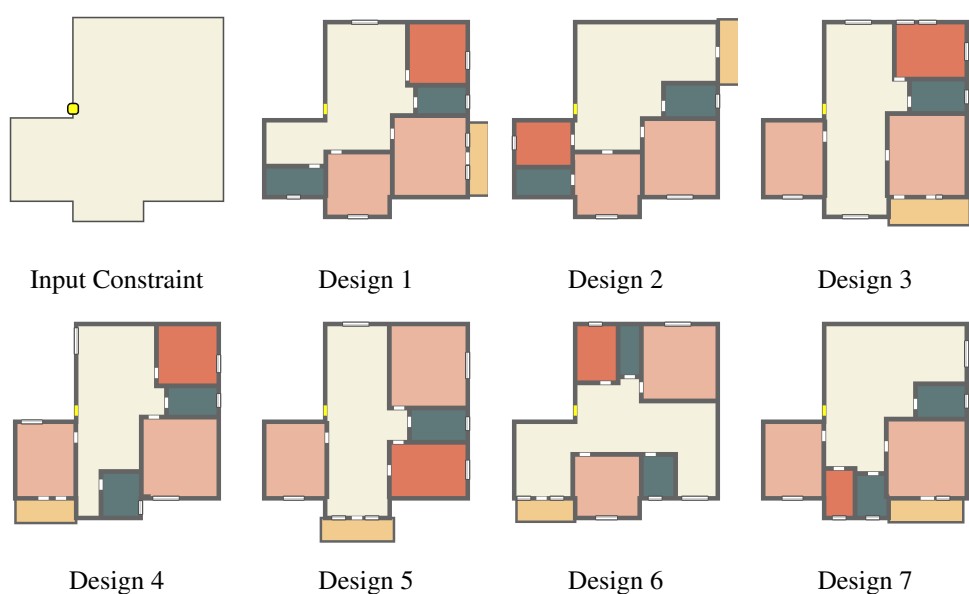

Input Constraint     Design 1     Design 2     Design 3

Design 4     Design 5     Design 6     Design 7

Figure 4: Here we used the same input constraint for all designs. By randomly selecting a blob from the predicted center channel in each case, we influenced the subsequent generation process, leading to different floor plan layouts. Each blob corresponds to a potential room center, and since every choice impacts the following steps, the model produces diverse designs. This demonstrates the flexibility of the generation process, as seen in the seven unique layouts displayed.

## 4 RESULTS

Although we used our own dataset to train GenPlan, we utilized the RPLAN dataset to conduct all experiments and comparisons to ensure a fair evaluation and demonstrate the generalizability of GenPlan. All experiments were conducted using an Intel Core i7-12700K CPU with 32 GB of RAM and an NVIDIA GTX 1050 GPU. For visualization tasks, we used Hunter (2007) and Garyfallidis et al. (2021)

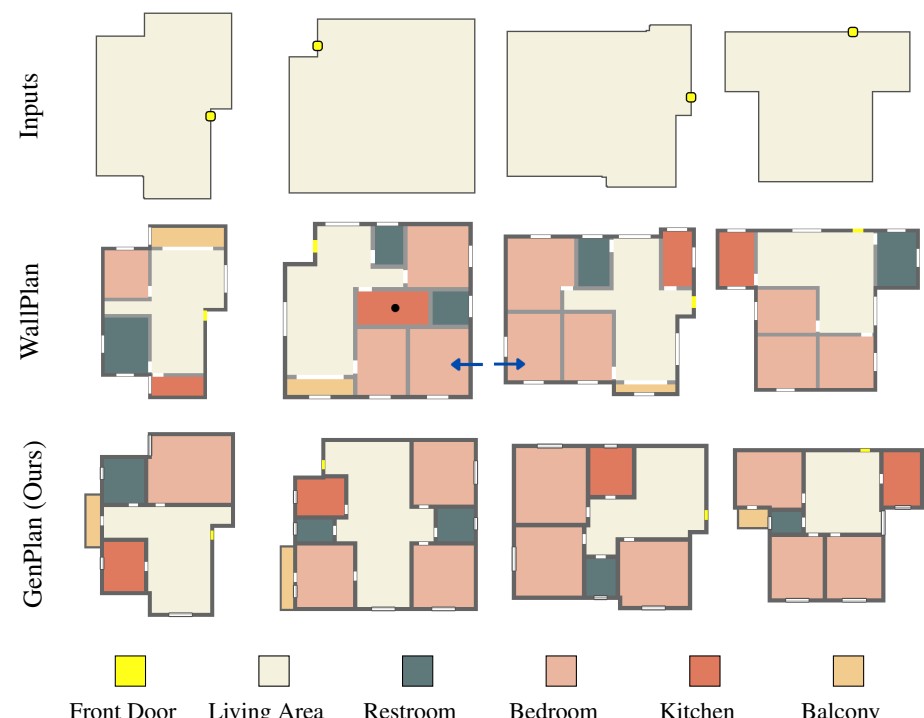

Figure 6: Performance comparison of GenPlan vs. WallPlan. Note that WallPlan generates trapped rooms (lacking entryways rooms annotated by the blue arrows) and places kitchens centrally (annotated by the black dot) without windows. Sun et al. (2022)

| Metric | WallPlan | GenPlan | Improvement |
|---|---|---|---|
| Trapped Rooms | 25 | 2 | 92% |
| Restrooms Without Outer Wall | 7 | 0 | 100% |
| Kitchen Without Outer Wall | 4 | 0 | 100% |
| Average Generation Time (s) | 9.75 | 3.35 | 65.6% |

Table 1: This table compares WallPlan and GenPlan across key metrics. GenPlan shows significant improvement, reducing trapped rooms from 25 to just 2 and entirely eliminating restrooms and kitchens without outer walls, unlike WallPlan. Additionally, GenPlan generates floor plans much faster, cutting the average time from 9.75 seconds to 3.35 seconds, a 65.6% speed improvement. These results highlight GenPlan's ability to create more functional and efficient designs with fewer structural issues.

| Bedrooms | WallPlan (%) | GenPlan (%) | Restrooms | WallPlan (%) | GenPlan (%) |
|---|---|---|---|---|---|
| No Bedrooms | 0.4% | 0.0% | No Restrooms | 0.0% | 0.0% |
| 1 Bedroom | 5.8% | 13.4% | 1 Restroom | 90.0% | 28.8% |
| 2 Bedrooms | 68.8% | 34.8% | 2 Restrooms | 9.8% | 40.8% |
| 3 Bedrooms | 23.4% | 42.4% | 3 Restrooms | 0.0% | 23.6% |
| 4 Bedrooms | 1.6% | 9.4% | 4 Restrooms | 0.0% | 6.8% |

Table 2: The two tables present a comparison between GenPlan and WallPlan, highlighting the diversity in the number of Bedrooms and Restrooms in the generated floor plans. This test demonstrates the variations in room counts across the two methods, with GenPlan showing greater flexibility, as it is not limited to a specific room count and can adapt to different design requirements

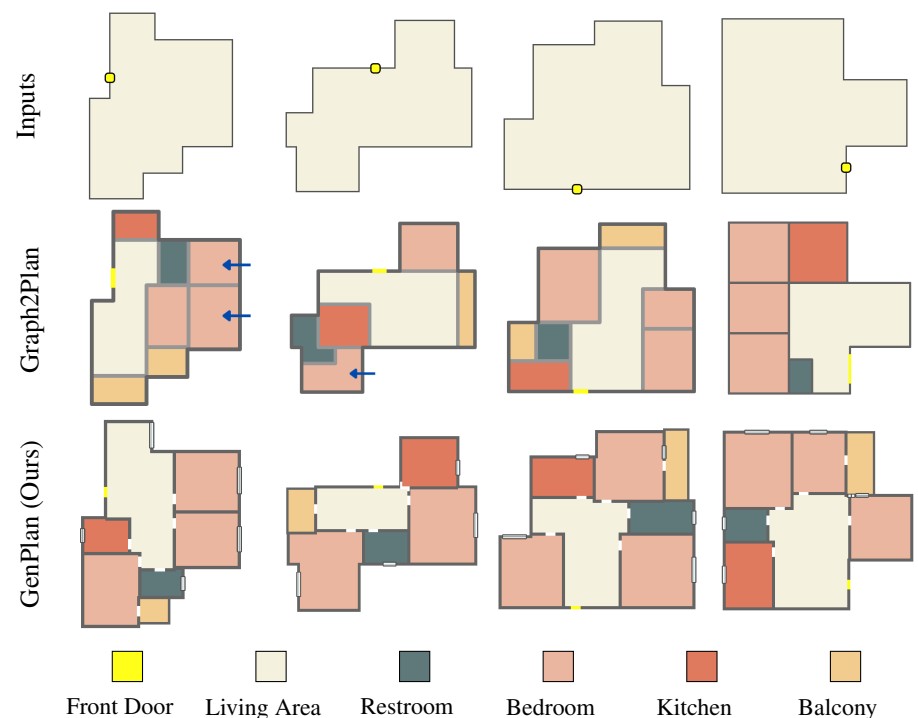

Figure 7: Performance comparison of GenPlan vs. Graph2Plan Hu et al. (2020). Note that Graph2Plan also generates trapped rooms (trapped bedrooms are annotated by the blue arrows). Additionally, some rooms do not share any outer walls, which our method never allows.

Tables 1 and 2 compare the WallPlan method with GenPlan (our method) across several metrics. Table 1, based on 100 floor plans, shows that GenPlan significantly reduces the number of trapped rooms from 25 to 4 (92% improvement) and eliminates windowless restrooms and kitchens (100% improvement for both), compared to 7 and 4 in WallPlan, respectively. Additionally, GenPlan is more efficient, with an average generation time of 3.35 seconds compared to WallPlan's 9.75 seconds, marking a 65.6% improvement. Table 2 provides a comparison of the predicted counts of bedrooms and restrooms between WallPlan and GenPlan across 500 floor plans. GenPlan demonstrates a more diverse generation of floor plans, with a more balanced distribution across different bedroom and restroom counts. While WallPlan predominantly generates floor plans with 2 bedrooms (68.8%) and 1 restroom (90.0%), GenPlan shows a higher percentage of floor plans with 3 bedrooms (42.4%) and 2 restrooms (40.8%). This diversity can lead to more functional and adaptable living spaces and simulation environments.

## 4.1 DISCUSSION

The GenPlan architecture significantly advances the field of automated architectural design by integrating convolutional and graph neural networks to enhance the generation and delineation of floor plans. This methodology not only accelerates the design process but also introduces precision in handling complex spatial relationships through the use of Transformer Convolution within the GNNs. This system offers a notable improvement over other published methods so far.

GenPlan has broad implications beyond architectural design. In urban planning, it can enhance the accuracy and efficiency of safety simulations and city layouts. Robotics applications could benefit from more precise environment mapping for better navigation and task performance. In the gaming industry, GenPlan can add realism and variety to game environments, improving player immersion. The film industry could use it to design set layouts and virtual environments more efficiently, potentially reducing production costs and time. These applications demonstrate GenPlan's potential to transform various industries by providing advanced design and simulation capabilities.

As future work, we plan to introduce additional dimensional constraints to make the system more capable of addressing real-life situations. This includes considering the orientation of the floor boundary and the geographical location to better utilize natural elements, such as sunlight for illumination and wind for cooling. We aim to achieve this by reverse engineering the ResPlan dataset to determine the optimal wind direction for each floor plan. The same applies to neighboring walls, as many complex floor plans arise from fitting multiple layouts within a single story, resulting in shared walls that cannot accommodate windows or doors. While ResPlan already contains initial data on neighboring walls, we are still in the process of refining this feature, so it has not been included in the current version.

## 4.2 CONCLUSION

In this work, we introduce GenPlan, a deep learning framework for the generation of realistic floor plans, tailored for use by architects, game designers, and developers. GenPlan harnesses the power of Transformer-based Graph Neural Networks (GNN) to enhance the precision of design outputs. The system is architected in a modular fashion, allowing for mid-process interaction to ensure the validity of the floor plan at each step prior to further progression. Moreover, GenPlan offers the ability to generate multiple valid designs for the same set of input constraints, representing a significant advancement in floor plan generation.

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
