# OpenReview forum: "GenPlan: Automated Floor Plan Generation"
_ICLR.cc/2025/Conference — Submitted to ICLR 2025_

### Official Review · Reviewer_A3iu · 2024-10-30

**Soundness:** 2
**Presentation:** 1
**Contribution:** 1
**Rating:** 3
**Confidence:** 4

**Summary:**

This paper proposes GenPlan, a method that converts the building boundary (i.e., the outmost wall boundary) with specified front-door location into a concrete floor plan. The framework consists of two main components: 1) GenCenter for predicting the room centers given the number of total rooms, and 2) GenLayout for generating the concrete room boundaries given the centers of rooms. GenPlan is trained on a self-collected dataset, but evaluated on RPLAN to compare against prior works. Quantitative and qualitative results show that GenPlan achieves higher success rate and more semantically reasonable floor plans compared to WallPlan.

**Strengths:**

1. The paper designs a reasonable two-stage generation framework for floorplans, which starts with predicting room centers given the number of rooms (GenCenter), then generates the per-room shape (boundary) given the room centers (GenLayout). GenCenter leverages ResNet and predicts room centers recurrently, while GenLayout leverages the Transformer Convolution architecture.

2. The paper collects a new dataset called ResPlan, which is claimed to be more diverse, realistic, and complex than the existing dataset RPLAN. This can potentially be a valuable contribution to the community.

**Weaknesses:**

1. There is a significant lack of related work in the paper. Only around 10 papers are cited, and a lot of highly relevant works in the literature of floorplan/geometry generation are not discussed at all or considered in results comparison. To be more concrete, RPLAN, Graph2Plan, WallPlan are mentioned or used as a baseline, but none of them are seriously reviewed in the related work; Highly-relevant works such as House-GAN and HouseDiffusion are not mentioned at all.

2. The method presentation is very sketchy and lacks details. All the equations are broken without serious mathematical definitions — the authors use plain texts to represent variables without providing any information about the dimensionality/shape of the variables. Solely based on the current description in Sec.3, it’s almost impossible to reproduce the work.

3. The paper claims a new dataset (ResPlan), which is more diverse, realistic, and complicated than the RPLAN dataset. There is no qualitative/quantitative comparisons between ResPlan and RPLAN to illustrate the superiority of the proposed dataset.

4. The quantitative evaluation metrics used in this paper are not standard. Neither perceptual user study results nor FID results are reported, making the quantitative comparison vague and not convincing.

5. Using bubble graphs as the condition is a common setting in previous works (House-GAN++, WallPlan, etc.), but this paper only conducts experiments using the building boundary as the condition, which does not fully present the capability of the proposed method.

6. The training/evaluation setting is confusing. It seems that the proposed method is trained with the new dataset (ResPlan) and the baselines are trained with RPLAN, and all methods are evaluated on RPLAN, leading to potential fairness issues.

**Questions:**

Please see the Weakness sections for the main problems of the paper.

Two more specific questions:
Following 5 of the Weaknesses section: Since the authors choose WallPlan as a main baseline, I don’t understand why the evaluation metrics used in WallPlan were not used in this paper.

Following 6 of the Weakness section: Why do different methods use different training datasets? Should the training data be the same to make sure the training settings are fair?

---

### Official Review · Reviewer_izQ9 · 2024-11-02

**Soundness:** 2
**Presentation:** 2
**Contribution:** 1
**Rating:** 3
**Confidence:** 5

**Summary:**

Paper presents a deep learning architecture for generating architectural floor plans. This architecture is built using two encoders and four specialized decoders to predict room locations and refine the predictions using a Transformer-based graph neural network (GNN).

**Strengths:**

1. Paper offers architects the ability to interact with the design system in real-time, allowing for on the fly adjustments.

2. The paper writing is clear and easy to follow.

3. The use of a Transformer-based Graph Neural Network (GNN) in the paper is reasonable and seems have solid theoretical background.

**Weaknesses:**

1. It has missed many important baselines, such as:

Hu, Ruizhen, et al. "Graph2plan: Learning floorplan generation from layout graphs." ACM Transactions on Graphics (TOG) 39.4 (2020): 118-1.

2. It has missed many important citations, such as:


Nauata, Nelson, et al. "House-gan++: Generative adversarial layout refinement network towards intelligent computational agent for professional architects." Proceedings of the IEEE/CVF Conference on Computer Vision and Pattern Recognition. 2021.

Shabani, Mohammad Amin, Sepidehsadat Hosseini, and Yasutaka Furukawa. "Housediffusion: Vector floorplan generation via a diffusion model with discrete and continuous denoising." Proceedings of the IEEE/CVF Conference on Computer Vision and Pattern Recognition. 2023.

3. it only handles Manhattan shapes, in real world scenario non-manhattan shapes happens a lot, and recent papers in floorplan generation such as Housediffusion already solve non Manhattan shapes.

Also you citation for housegan++ is wrong
 you have cited as: " Naoki Nauata, Seyed Hamid Hosseini, Kai-Hung Chang, Hang Chu, Chieh-Yi Cheng, and Yasutaka Furukawa. House-gan++: Generative adversarial layout refinement network towards intelligent computational agent for professional architects. In Proceedings of the IEEE/CVF Conference on Computer Vision and Pattern Recognition (CVPR), pp. 13627–13636, 2021. doi: 10.1109/ CVPR46437.2021.01342."
While the correct citation is : "Nauata, Nelson, Sepidehsadat Hosseini, Kai-Hung Chang, Hang Chu, Chin-Yi Cheng, and Yasutaka Furukawa. "House-gan++: Generative adversarial layout refinement network towards intelligent computational agent for professional architects." In Proceedings of the IEEE/CVF Conference on Computer Vision and Pattern Recognition, pp. 13632-13641. 2021."
Having different authors names for same paper is very alarming mistake.

**Questions:**

Have you calculated metrics such as graph distance or Realism (using human evaluators)?

---

### Official Review · Reviewer_wSAj · 2024-11-04

**Soundness:** 2
**Presentation:** 3
**Contribution:** 1
**Rating:** 3
**Confidence:** 4

**Summary:**

The paper provides an approach to generate floor plans based on input constraints provided. The key contributions are:
1. An optional step to determing the number of rooms in a floor area
2. A ResNet model based approach for deciding the center of the rooms
3. A GAN based approach for deciding the perimeter of each room.

A new dataset was created containing 17000 images with floor plans scraped from different realtor website. These images were preprocessed and labelled for training the model.

The ResNet model and the GAN models are trained independently in two different phases

The proposed approach is tested against 100 floor plans in the RPLAN dataset and the proposed approach performs better than the baseline approach WallPlan

**Strengths:**

1. The paper is well written. It is easy to read and follow the paper. The visualizations are supportive of the paper description.
2. The problem being addressed in this paper is an important and a practically useful problem.

**Weaknesses:**

Lack of innovation:
1. The paper proposes a less technically sounds approach for floor plan generation.
2. The paper uses 2-3 already existing pretrained models/approaches to generate floor plans. LIttle innovation is performed on top. While I read the paper, I did not learn anything new.

Technical choices not explained:
1. The paper uses ResNet18, ResNet 101, and a GNN based transformer implemented in Fey & Lenssen (2019) in the main pipeline. The reason behind the choices of this specific architecture is not described anywhere.
2. No ablation studies conducted on models, architectures, hyperparameters etc. It is difficult to scientifically accept this solution

Weak experimentation:
1. The proposed ResPlan dataset is purely used for training purposes. For testing/evaluation a much smaller dataset RPLAN with only 100 images is used. The experiments and comparison are not strong with such a small dataset.
2. Even in the proposed approach, there could be a lot of conflicts while generating the floor plans. The experiments do not talk about how they are handled?
3. The hyperparameters are ad-hoc and the impact of them on the results are not studied.


Poor comparison with the literature:
1. The proposed approach is compared with only WallPlan paper in literature. While there are many other approaches discussed in the literature review section, there is no experimental comparison against them.

**Questions:**

1. Why only the selected architecture (ResNet18, ResNet 101, and a GNN based transformer implemented in Fey & Lenssen (2019)) were chosen? Why not any other architecture?

2. Need additional experimental comparison with other approaches discussed in literature.

3. Suggest to split the ResPlan dataset into training and testing and show the results of the proposed approach in the test dataset.

4. Need additional clarity, including expeirmental demonstration, on how the conflict is addressed during generation - for ex, conflicts in center generation, or conflict during perimeter generation.

5. Introduction claims that the proposed approach provides better flexibility as compared to other approach - that is not showed experimentally. The only part that is shown is that the proposed methods is better at generating floorplans with more than one bathroom. That looks ad-hoc.

---

### Official Review · Reviewer_SfR9 · 2024-11-05

**Soundness:** 3
**Presentation:** 3
**Contribution:** 3
**Rating:** 8
**Confidence:** 4

**Summary:**

This paper presents a method for generating interior floorplan layouts (and exterior balconies) for a one-floor building given the exterior shape and the location of the entrance door, and optionally a specification of the number of bedrooms and/or bathrooms to include in the layout.

The approach uses a two stage system; first room centers are predicted recurrently using a CNN trained to output images highlighting the room centers given image-encoded data about the floorplan layout and previously chosen room centers.

In the second stage a GNN is used to regress the coordinates of the room corners given an initialization of room centers, exterior walls, and a topology connecting the interior rooms with each other and their nearest exterior walls.

This work also contributes a new dataset (ResPlan) of 17k floor plans sourced from RealEstate websites that claims to be more diverse (in number of bedrooms and bathrooms at least) than existing research datasets.

**Strengths:**

This paper takes a clever approach to solving a combined discrete-continuous generation problem. The algorithm details are mostly well described (aside from some typos and ordering issues described below that could easily be rectified with a clear overview and some editing), and the results look qualitatively good.

**Weaknesses:**

The evaluation metrics used to compare GenPlan against the WallPlan baseline are weak and somewhat arbitrary. Fewer trapped rooms and faster generation seem unambiguously good, but interior bathrooms and kitchens do exist in real buildings (especially small interior powder rooms in en-suites, etc.). While I would say that it is usually more desirable to have windows in a room than not, I'm unconvinced that this local improvement is necessarily part of a global-optimum for room design. For these, and the room diversity metrics in Table 2, I would guess the difference is partly or largely attributable to difference in training data distribution between the methods. It would be nice to compare against a version of the WallPlan algorithm trained over the same data set, or to compare the methods to similar metrics in their training set's distribution.

Interior rooms other than the catch-all "living area" can only be axis-aligned rectangles (up-to the external wall shape)

The Related Work section is very sparse. It would be helpful if it at least discussed the RPlan dataset and WallPlan, and situated this paper's dataset and method in relation to them specifically, since those are the primary dataset and baseline used for evaluation.

The exposition of the method is difficult to follow in places owing to variables and parts of the method being referenced before the are properly introduced. For example, section 3 starts with "Once a room center is predicted" then jumps straight into a description of the GNN, and does not even mention the room count prediction step. The methodology section is clear on a second reason once the context has been gained, but it would be much better to have a clear and complete overview of the whole method at the beginning of section 3. I would move Figure 3 to the beginning of section 3 and reference it in this overview, add a more detailed caption to figure 3, as well as images of the graphs used as GNN input (since they have an obvious spatial embedding) to make their structure clear.

**Questions:**

What, specifically, is meant by "architecturally sound" (in refence to Figure 4, L215).

In the floorplan diagrams in Figures 1, 4, and 5 there are many small artifacts in the walls -- sections of wall or corners that appear to be jutting into the rooms a bit. For example, see the right-wall of the top-left room in Figure 1 (a). Do these inconsistencies truly exist in the generated results, or are the artifacts a result of the figure-production process?

Also in floorplan diagrams; the distinction between windows and doors in very hard to see, consider using a heavier line-weight in the windows.

Figure 1(b) is missing the windows indicated by Figure 1(a)

Why are the doors all different sizes, and how is that determined?

Why does Figure 5 appear before Figure 4, and are are their references in the main text correct?

L142: Should N_{counts} actually be N_{rooms} in Eq(1) (like on L144)?

Should F_{shared} in Eq (2) (L161) actually be F_{layout}? F_{shared} is also defined in equation (4)

I would suggest moving Eq (4) up to L180 to be closer to the text that describes it.

How do the properties of Graph transformer convolutions (L256-269) relate to the claim of "advanced capabilities in handling spatial data" (L254), or is there a reference missing that demonstrates graph transformer convolutions are better for spatial data?

L256-259 claims that self-attention allows for global-scale processing; how does a global receptive field follow from local attention?

In general, the claims about why a graph transformer were used  appear to me to be more educated speculation that motivated the choice of convolution rather than tested hypotheses. I would either remove them, or tone down the language a bit to indicate that these are suspected rather than proven advantage (unless there are external references or ablation studies to validate the claims).

Can you cite and/or describe the Laplacian of Gaussian blob detection technique?

L169 -- how exactly are the room center images transformed int 8x8x512 feature maps (required for reproducibility)

L195; what objective was used for the initial training of the Layout Encoder?

How is the order to run the different Decoder types chosen?

L302-305 -- How is an architecture used to predict point centers in 2D space adapted to predict contiguous regions constrained to room boundaries and exterior walls?

L127: What encoding is used for coordinates in the GNN? Are these regressed floats or categorization over a discretized space (e.g. pixel positions)?

L138: What is the source of the "historical data" -- is this referring to ResPlan?

L480-482: How would GenPlan be used to enhance accuracy and efficiency of safety simulations in urban planning, or environment mapping for robotics? It is a generative system, and these tasks appear to be analytic. The claims for game and film application seem much more plausible.

Figure 6: the WallPlan example in the fourth column appears to have a trapped room in its bottom-left corner but this is not indicated with an arrow. Is there a way in which this room is not trapped?

L321: What does "efficiently integrated" mean?

L315: "Geometric Buffering" is described as a key technique, but the algorithm is not given, just a description of its outputs. How exactly does it work (required for reproducibility).

---

### Official Review · Reviewer_uRnE · 2024-11-07

**Soundness:** 1
**Presentation:** 1
**Contribution:** 3
**Rating:** 3
**Confidence:** 4

**Summary:**

The paper introduces a new 2-stage method for generating floorplans based on an initial boundary. The method uses a CNN to predict room centers and a GNN to predict room rectangles based on the centers.

**Strengths:**

- **S.1:** The paper proposes an interesting method. I think the center -> outline approach has a lot of merit.

**Weaknesses:**

- **W.1:** The writing is not great. The method is unclear and hard to follow. The network diagram (Fig.3) could be easier to parse and should be much earlier in the paper - ideally on the top of page 2. There are also tons of details missing, like how many layers does the CNN have, what are these layers, how many layers does the GNN have, etc. Also, in general the method section could benefit from more examples, especially part 3.1 - how the data is fed into the encoders. Also, someone needs to do a pass of proofreading.
- **W.2:** The method imposes weird, unpractical restrictions - most rooms can only be rectangular and living rooms are always the hallways that connect the other rooms. I don't think that's based on any architectural consensus. That makes me question the usefulness of this method.
- **W.3:** The authors created a new dataset and write about that being a new "standard for comparison in this field" and then don't make the dataset public and don't release any code with the submission. Also, the method is pretrained on their proprietary dataset and then evaluated on the public RPLAN dataset, which makes it impossible to disentangle the results as belonging to the dataset or belonging to the method. The dataset needs to be included in the submission or even better, publically hosted on GitHub, or else the method needs to be trained AND evaluated on RPLAN, otherwise the results are meaningless.
- **W.4:** The illustrations are confusing and the tables misleading.
  - Fig.1: Why is there a 3D visualization? Does your method have anything to do with 3D visualizations? Where are the doors and windows in the 3D visualization? Is this just eye candy?
  - Fig.2: What is the outline of the house/apartment? I can't evaluate the goodness of these suggestions without a frame of reference.
  - Fig.3: is okay but it'd be nice to see examples of the actual data that goes into the different encoders, possibly in a separate figure (see W.1)
  - Fig.5: the bedroom and balcony colors are indistinguishable. Also, is there always a balcony added? Is that architecturally possible?
  - Fig.4: why is Fig.4 underneath Fig.5? Why is there a variable number of washrooms? I thought your method predicts a number of washrooms+bedrooms, so that number should always be the same, no?
  - Fig.6: how is this different from Fig.7?
  - Tab.1: the absolute numbers are meaningless. This should be percentages of the dataset. And the "Improvement percentage" column is deceptive and should be removed.
  - Tab.2: I thought that Gen Plan can generate a flexible number of bed+bathrooms. Why are there summary statistics? I thought the number of bath+bedrooms is an input parameter to the GNN.

**Questions:**

- **Q.1:** Around line 50, you talk about deep learning having issues achieving a golden ratio. Does your method achieve golden ratios?
- **Q.2:** How do architects interact with your system other than changing the room number?
- **Q.3:** Are window positions not constrained to certain sides of the building?
- **Q.4:** Was your dataset collected with the consent of all the different real estate websites? Where can I find more information about that?
- **Q.5:** Line 138... what's "historical data"?
- **Q.6:** line 154 "desigred", line 156 "roos"... there are many of these. Please proof-read a paper before submitting!
- **Q.7:** line 167 you mean "predicted" instead of "detected"?
- **Q.8:** line 377 - you can just say "Matplotlib" rather than "Hunter (2007)"

**Details Of Ethics Concerns:**

Dataset possibly illegally scraped. Unclear.

---

### Official Review · Reviewer_c3AT · 2024-11-08

**Soundness:** 2
**Presentation:** 2
**Contribution:** 1
**Rating:** 1
**Confidence:** 5

**Summary:**

This paper presents a graph transformer-based framework to generated interior floorplans from a given input boundary in an end-to-end manner. Room center segmentation heat map are first predicted, then a blob detection algorithm is employed to process the segmentation and generate wall locations. Finally GNNs will aggregate multi-room information and optimize the holistic layout.

**Strengths:**

The paper presents a feasible system for end-to-end floorplan generation.

**Weaknesses:**

1. The proposed system lacks of technical novelty. The process of room center segmentation prediction then vectorization is similar to the wall graph generation phase in Wallplan. The network architectures (CNNs, Recurrent networks, Transformer-based GNNs) are not technically interesting as well.
2. The paper lacks a comprehensive and fair comparison with state-of-the-arts. Specifically, quantitative metrics like FID and GED (graph distance) as well as real user study to measure the generated floorplans are required.
3. The paper misses some important literature views, such as the latest process in floorplan generation topic, as well as the application of GNNs in computer vision and generative models.
4. There is no ablation study to validate any contribution of the proposed system.

**Questions:**

Based on the notable issues in technical novelty, lack of completeness of experiments, and the missing literature reviews in this paper, I think this paper have not reached the bar to a top conference like ICLR. Please revise it carefully and fix these major drawbacks.

---

### Meta-Review · Area_Chair_fiYT · 2024-12-17

**Metareview:**

After careful consideration of the six expert reviews and noting the absence of author rebuttal, I recommend rejecting this submission. While the paper presents an interesting approach to automated floor plan generation using a combination of CNNs and Transformer-based GNNs, there are several fundamental issues that prevent it from meeting ICLR's acceptance criteria.

The paper's primary technical contribution centers on a two-stage generation framework: GenCenter for predicting room centers and GenLayout for generating room boundaries. While this approach is conceptually reasonable, the reviewers identified significant gaps in both technical depth and experimental validation. The method presentation lacks crucial implementation details, which make reproducibility nearly impossible. Mathematical formulations are inappropriately defined, with variables presented without proper dimensionality or shape specifications.

A major concern raised by multiple reviewers is the paper's inadequate treatment of related work. Despite the extensive literature in floor plan generation, the paper cites few references and omits discussion of several highly relevant works, including House-GAN, HouseDiffusion, and Graph2Plan. Even for the works that are cited, such as RPLAN and WallPlan, the discussion lacks depth and proper context. One reviewer also noted an alarming error in citation, where author names were incorrectly listed for a key reference.

The experimental evaluation raises serious concerns about fairness and comprehensiveness. The authors train their model on their proprietary ResPlan dataset while comparing against baselines trained on RPLAN, creating an uneven comparison. The evaluation metrics used are non-standard, lacking both perceptual user studies and established metrics like FID scores. In addition, while the paper claims ResPlan is more diverse and realistic than existing datasets, it provides no quantitative or qualitative evidence to support this assertion.

The technical limitations of the approach are also significant. The method only handles Manhattan shapes, a constraint that limits its practical applicability, as real-world scenarios often involve non-Manhattan geometries. The paper also restricts its evaluation to building boundary conditions, neglecting to demonstrate capability with bubble graphs, which are common in previous works.

These issues are compounded by the absence of crucial components that would strengthen the work: ablation studies to justify architectural choices, detailed explanations of conflict resolution in the generation process, and proper comparison with state-of-the-art methods using standard metrics. The lack of author response to these concerns further suggests that these fundamental issues cannot be readily addressed.

While the paper addresses a problem in architectural design automation and presents some interesting ideas, the combination of technical shortcomings, inadequate literature review, and experimental limitations make it unsuitable for publication at ICLR 2025. For future submissions, the authors are recommended to focus on providing more rigorous technical details, comprehensive comparisons with existing work, and thorough experimental validation using standard metrics.

**Additional Comments On Reviewer Discussion:**

The authors did not provide any rebuttal. None of the issues raised by the reviewers were discussed and addressed.

---

### Decision · Program_Chairs · 2025-01-22

Reject